# Tropism of the Novel AAVBR1 Capsid Following Subretinal Delivery

**DOI:** 10.3390/ijms23147738

**Published:** 2022-07-13

**Authors:** Lara Carroll, Hironori Uehara, Xiaohui Zhang, Balamurali Ambati

**Affiliations:** 1Moran Eye Center, 65 Mario Capecchi Way, Salt Lake City, UT 84132, USA; 2Phil and Penny Knight Campus for Accelerating Scientific Impact, 6231 University of Oregon, Eugene, OR 97403, USA; huehara@uoregon.edu (H.U.); xiaohuiz@uoregon.edu (X.Z.); bambati@uoregon.edu (B.A.)

**Keywords:** subretinal delivery, adeno-associated virus (AAV), gene therapy, transduction, binding motif

## Abstract

A serious limitation of current adeno-associated viral (AAV) capsids employed for subretinal delivery is achieving adequate lateral spread beyond the injection site, required for the efficient delivery of gene therapy to the outer retina and/or RPE. AAVBR1 is a unique AAV with exceptional tropism for CNS microvasculature following systemic delivery. Here, we used in vivo and ex vivo analysis to show that subretinal delivery of AAVBR1.GFP in mice achieves superior tropism to RPE and outer retina than either AAV2.GFP or AAV8.GFP, two of the most common capsids used for subretinal delivery. At a low (5 × 10^8^ vg) subretinal dose, the AAVBR1.GFP signal was visible by 48 h and significantly surpassed peak fluorescence of other AAVs in retina and RPE. The co-injection of AAVBR1.GFP with the AAVBR1-specific heptapeptide, NRGTEWD, significantly blocked the AAVBR1.GFP signal, but had no effect on AAV2.GFP fluorescence, confirming that AAVBR1’s enhanced tropism for RPE and outer retina derives from this 7AA modification within the capsid-binding motif. Enhanced dispersal and consequent transduction suggest that AAVBR1 can be employed at a lower dosage than the standard AAV2 capsid to achieve equivalent expression for gene therapy, warranting further evaluation of its utility as a therapeutic vehicle for subretinal delivery.

## 1. Introduction

The unequivocal value of AAV-based gene therapy for the outer retina was cemented by the restoration of sight to patients with previously incurable photoreceptor degeneration in Leber’s congenital amaurosis [1]. AAVs present the appealing prospect of long-term delivery of gene therapy, circumventing the risks and high patient burden associated with frequent injections. Meanwhile, subretinal injection places therapy into direct contact with RPE cells and photoreceptors, making it an excellent site for drug delivery in patients with vision-threatening disorders of the outer retina. However, achieving adequate lateral spread of AAVs across the subretinal space at physiologically safe doses is an ongoing challenge, as viral load tends to remain close to the subretinal bleb created during injection [2]. This is particularly true for AAV2, which remains the best characterized and only AAV capsid currently approved for human ocular gene therapy. Although intravitreal injection is more clinically straightforward than subretinal injection, available capsids capable of penetrating the inner limiting membrane and diffusing to the outer retina have had limited success in preclinical and clinical trials, and the higher doses required to offset vitreal dilution increase the potential for AAV toxicity and/or adverse immune responses [3,4,5]. The recently engineered AAVBR1 is a capsid differing from AAV2 by seven consecutive amino acids in its binding motif. Developed via artificial selection in mice to target CNS microvascular endothelium, the systemic delivery of AAVBR1 shows excellent specificity and expression in mouse brain vasculature at a moderate dose of 5 × 10^10^ vg in 100 μL [6], while systemic injection of 1 × 10^11^ vg in 100 μL showed excellent signal in retinal microvasculature [7]. While examining its retinal tropism, we fortuitously found that the moderate dosage of AAVBR1 (5 × 10^8^ vg) showed exceptional lateral spread following subretinal injection, with strong tropism for both RPE and the outer retina. Here, we evaluated the potential of AAVBR1 for direct RPE/photoreceptor targeting in comparison to two AAVs commonly employed for subretinal delivery in preclinical studies: AAV2 and AAV8.

## 2. Materials and Methods

### 2.1. Animals

This study was approved by the University of Utah Institutional Animal Care and Use Committee (IACUC), protocol number 18−03007. C57Bl/6J mice were originally purchased from Jackson Labs and bred in our facility. All mice (both males and females) were between 8 and 12 weeks old at the time of injection and comparisons were always performed on mice injected on the same day (i.e., no inter-experimental comparisons). Two mice each were used to evaluate AAVBR1.GFP expression following intravitreal, systemic (tail vein or retro-orbital), and subretinal AAV delivery. Two additional mice received retro-orbital injections of DPBS to serve as controls for systemically injected AAVBR1.GFP. In vivo expression time course was compared among AAV2.GFP, AAV8.GFP, and AAVBR1.GFP (2 mice each) following bilateral subretinal injection, and eyes were embedded for sectioning at the 6-week endpoint. Additional mice were subretinally injected with AAV2.GFP (5 mice), AAV8.GFP (6 mice), and AABR1.GFP (5 mice) to compare RPE and corresponding retinal flatmounts at the 6-week endpoints. A final experiment examined 10-week expression endpoints following subretinal injection of AAV2.GFP (2 mice), AAVBR1.GFP (3 mice), or each vector co-injected with a BR1-specific blocking peptide (2 and 3 mice, respectively). For analysis, each injected eye was considered statistically independent. One AAV2.GFP-injected mouse died before the experimental endpoint. The final number of eyes used for each statistical analysis is indicated in the figures.

### 2.2. Injections

Tail vein injections were performed with 100 μL 5 × 10^8^ vg/μL (5 × 10^10^ vg total) in alert, restrained mice. Ocular injections (intravitreal and subretinal) were performed on mice anesthetized with 90 μg/10 mg/kg body weight ketamine/xylazine). Prior to ocular injection, eyes were dilated with a 1% topical tropicamide solution (to visualize needle position and subretinal blebs) and anesthetized with topical proparacaine. A sterile 30-gauge needle was used to puncture a clean hole approximately 1 mm below the limbus, followed by the insertion of a 33-gauge blunt tip Hamilton microsyringe (Hamilton Company, Reno, NV) containing the appropriate AAV. Intravitreal injections were released approximately 1–2 mm beyond the puncture into the vitreous. For subretinal injections, the needle was advanced beyond the opposing retinal wall to release syringe contents at the apical RPE surface. All intravitreal and subretinal injections contained 5 × 10^8^ vg in 1 μL volumes. Topical erythromycin was applied after injection prior to returning mice to home cages for recovery. Subretinal blebs were confirmed immediately following injection by direct fundus visualization with a coverslip under a dissecting scope. Although retinal hemorrhages are not uncommon during subretinal injection in mice, eyes were only excluded from analysis if a hemorrhage obscured our ability to confirm the subretinal bleb.

### 2.3. AAVs and Blocking Peptide

AAVBR1 capsid plasmid DNA was kindly provided by J. Korbelin, amplified in DH5α cells, and purified with Qiagen DNA Maxi kits in our facility. The GFP expression construct, pAAV.CMV.AcGFP, was cloned in our laboratory using AcGFP cDNA from the pIRES2-AcGFP1 plasmid (Clontech Laboratories, Mountain View, CA, USA) and inserted into the pAAV-MCS backbone (Agilent Technologies, Santa Clara, CA, USA). Packaging of our GFP construct into AAVBR1, AAV2 and AAV8 capsids was conducted by Vigene Biosciences (Rockville, MD, USA). Viral stock for the three GFP AAVs was aliquoted and stored at −80 °C. Immediately prior to injections, viral stock was thawed and diluted in PBS to 5 × 10^8^ vg/μL, or to 1 × 10^9^ vg/μL for blocking peptide co-injection studies (at a 1:500 molar ratio of virus:peptide). The 9AA blocking peptide sequence, GNRGTEWDA, contained the 7AA unique AAVBR1 sequence plus an additional flanking AA from the AAV2 capsid sequence on either side (5′-G and 3′-A) to configure the ends. The peptide was purchased from Genscript Biotech (Piscataway, NJ, USA), diluted in PBS, aliquoted and stored at −80 °C.

### 2.4. Imaging and Analysis

A Heidelberg Spectralis ophthalmic imaging platform was used to capture black and white images of in vivo fundus GFP fluorescence. For ex vivo imaging, fields were captured of retina/RPE flatmounts or sections at 10× or 20× objective, respectively, with an EVOS FL imager. For flatmount retinas or RPEs, 10× objective stitched images of the entire tissue were acquired with the EVOS using equivalent fluorescence parameter settings. Fluorescence (% fluorescence area) was quantified in images of stitched flatmounts imported into Fiji [8], converted to 8-bit, and inverted so that fluorescent pixels appeared dark. Areas of retinas or RPEs were outlined with the ImageJ polygon tool and pixel values were converted to binary states by setting equivalent minimum and maximum threshold values for each image. The total number of black and white pixels enclosed by the polygon was determined using the histogram tool, and the % fluorescence area was calculated as #black/(#black + #white) pixels. Statistical analysis was performed using unpaired *t*-tests or Kruskal–Wallis tests (non-parametric one-way ANOVA) with Dunn’s test for multiple comparisons in GraphPad Prism (V 9.3.1).

### 2.5. Fluorescent Immunohistochemistry: Brains, Flatmounts, and Sections

Brains were harvested from mice anesthetized with ketamine/xylazine. After opening the chest cavity, the right ventricle was punctured and the animal was perfused with 30 mL of ice-cold PBS followed by 30 mL of ice-cold 4% paraformaldehyde. Eyes from these animals were harvested for flatmounts and brains were gently dissected from the skull cavity, bisected, and placed in 4% paraformaldehyde overnight before washing and embedding in agar. A Leica VT100S vibratome was used to create 70 μm brain slices. Floating slices were processed for immunofluorescence by first permeabilizing them for 6 h in 0.5% TritonX/PBS at 25 °C, followed by several washes in PBST (PBS with 0.1% PBS), before then being blocked for 1 h at 25 °C in 10% NGS/PBST. Sections were incubated overnight at 4 °C in 1:50 chicken anti-GFP in PBST to amplify the AAVBR1.GFP signal (GFP−1020; Aves Labs, Davis, CA, USA). The next day, the unbound antibody was removed in six 15 min PBST washes before incubating sections in 1:200 Alexa Fluor^TM^ 488 goat anti-chicken (Invitrogen, Waltham, MA, USA) with 1:100 Alexa Fluor^TM^ 647 GS-IB_4_ (Invitrogen) to label blood vessels. The GS-IB_4_ signal was pseudo-colored red to help visualize colocalized vessels. All other mice were euthanized by CO_2_ asphyxiation for harvesting globes. Flatmounts were processed by first puncturing the cornea before fixing the globe in 4% PFA for 1 h at 4 °C. After removing the anterior segment, RPE/choroid and retinas were washed in PBS and gently separated for flatmounting or fixed for an additional hour at 4 °C to embed for sections. RPE/choroid tissue was further processed to cut away all muscle adhering to the sclera before flattening the cup with 6–8 radial cuts. Tissues were mounted flat on slides and immediately coverslipped using Fluoromount-G mounting medium (Thermo Fisher). The tissue for sectioning was equilibrated through 15% and 30% sucrose solutions (in PBS), embedded in Tissue-Tek^®^ O.C.T. freezing compound (Sakura Finetek, Torrance, CA, USA), and sectioned on a microtome at 10μm. Sections were washed in PBS and coverslipped using Fluoromount-G.

## 3. Results

### 3.1. Variable Ocular Expression of AAVBR1.GFP Using Different Delivery Routes

We first sought to characterize the retinal expression of AAVBR1.GFP using three different delivery routes: systemic (retro-orbital or tail vein, 5 × 10^10^ vg in 100 μL, N = 2 each), intravitreal, or subretinal (bilateral injections of 5 × 10^8^ vg in 1 μL, N = 2 each). These doses are based on the original systemic study of Korbelin et al. (2016), which found high targeting efficiency in mouse CNS (brain and spinal cord) microvasculature with 100 μL systemic delivery of 5 × 10^8^ vg/μL AAVBR1. One month after injection, fundus images showed only rare punctate signal from intravitreal and systemic injections; however, the subretinal delivery of AAVBR1 gave strong signal both near and distant to the injection site (Figure 1A). As our expression vector employs the CMV promoter rather than the CAG promoter employed in the originally published reagent [6], we examined brains of systemically injected mice to confirm that CMV worked as expected and could reliably target brain microvascular endothelium. Perfusion-fixed brains of mice retro-orbitally injected with either AAVBR1.GFP or PBS were sectioned and processed for immunofluorescence to co-localize the fluorescent GFP signal with vascular marker GS-IB_4_. As expected, brains of control-injected mice showed only background autofluorescence and no GS-IB_4_ colocalization, while AAVBR1.GFP-injected mice showed exceptional specificity for brain vasculature (Figure 1B) with occasional signal in neuronal cells (data not shown). An extremely rare signal was found in the retinal microvasculature of these mice (Figure 1C), suggesting that higher doses and/or a stronger promoter are required to target retinal vascular endothelium, as was shown by Ivanova et al. (2021).

### 3.2. Subretinal Injections: Time Course of Signal Expression and Transduction Efficiency

To compare the transduction time course and tropism via subretinal administration of AAVs, 5 × 10^8^ vg of AAVBR1.GFP, AAV2.GFP, or AAV8.GFP was injected subretinally into both eyes. At six weeks post-injection, GFP expression was quantified ex vivo in retinal and RPE flatmounts and confirmed in sections. While the in vivo fundus GFP signal was visible in all AAV-injected eyes by two weeks, AAVBR1 and AAAV8 (but not AAV2) expressed GFP as early as 48 h post-injection. Despite similar early onset of AAVBR1 and AAV8 expression, fundus images showed higher fluorescence from AAVBR1 injection than from AAV2 and AAV8 injections at the two-week timepoint, which clearly extended beyond the injection site to all ocular quadrants at both two and four weeks in AAVBR1-injected eyes. In contrast, the signal from AAV2 and AAV8 injections remained close to the injection site. Figure 2 shows three representative eyes imaged at 48 h, 2 weeks and 4 weeks.

Consistent with in vivo fundus images, ex vivo analysis of retina and RPE at the 6-week endpoint showed that the GFP signal occupied a significantly greater area of AAVBR1-injected RPE and retinal flatmounts than the GFP signal did in either AAV2- or AAV8-injected eyes (Figure 3A). The fluorescent area of AAVBR1-injected eyes was approximately fivefold greater than that of AAV2 and AAV8 in the RPE (AAV2 and AAV8 vs. AAVBR1, both *p* < 0.001; Kruskal–Wallis test followed by Dunn’s multiple comparisons test). The retinal GFP signal from AAVBR1 injection was 10-fold greater than that of AAV2 and >two-fold greater than that of AAV8 (AAV2 vs. AAVBR1, *p* < 0.001; AAV8 vs. AAVBR1, *p* = 0.037; AAV2 vs. AAV8, n.s. Kruskal–Wallis test followed by Dunn’s multiple comparisons test). Images of all AAV.GFP-injected RPE/retinae used in this analysis can be viewed in Appendix A. An examination of retinal cross-sections at the six-week endpoint corroborated the finding that even in fields where the GFP signal was strong in the underlying RPE of all three groups, more GFP-positive cells could be found in the overlying retina of AAVBR1-injected eyes than in either AAV2- or AAV8-injected eyes (Figure 3B). However, within the retina, GFP-positive cells were found almost exclusively in the outer nuclear layer of AAVBR1- and AAV8-injected eyes, whereas sparse fluorescent cells could be seen in the inner nuclear layer (INL) and occasionally the ganglion cell layer (GCL) of AAV2.GFP-injected eyes.

### 3.3. Co-Injection of the Unique AAVBR1 Peptide Motif Is Sufficient to Block AAVBR1 Expression

The AAVBR1 capsid differs from AAV2 by only seven consecutive amino acids (NRGTEWD) in its binding motif. To determine whether this peptide would compete specifically with AAVBR1 for RPE and/or retinal cell-surface targets and limit AAVBR1 transduction/expression, we subretinally co-injected the purified AAVBR1-specific peptide with either 1 × 10^9^ vg of AAVBR1.GFP or AAV2.GFP (1:500 molar ratio virus:peptide) and measured the flatmount expression compared to corresponding AAV.GFPs injected with vehicle alone. The fluorescence quantification of flatmounted tissues 10 weeks after injection indicated that peptide co-injection did not alter AAV2 capsid binding in either RPE or retina (Figure 4), while it significantly abrogated the RPE signal following AAV-BR1.GFP/peptide co-injection (*p* < 0.005, Student’s two-tailed *t*-test). In retina, AAV-BR1.GFP showed a similar trend of signal loss with co-injected peptide (*p* = 0.06, Student’s two-tailed *t*-test). Images of all AAV.GFP-injected RPE/retinae (with and without blocking peptide) used in this analysis can be viewed in Appendix A.

## 4. Discussion

AAV-based ocular gene therapy can be sustained for years with a single injection, substantially reducing patient burden and the ophthalmologic risk associated with repeated ocular needle penetration. Subretinal injection is currently the best AAV delivery option for inherited diseases of the outer retina. Although intravitreal injection might be more practical in terms of clinical simplicity, ongoing efforts to engineer intravitreally injected AAVs for improved tropism to the outer retina have failed to perform as hoped, often requiring higher doses to disseminate virus to target sites, with potential for increasing AAV toxicity and adverse immune responses [9]. Improvements in needle design, particularly the introduction of microneedles that can target the subretinal space with trans-scleral injection, are expected to revolutionize ophthalmic disease management for the outer retina [10].

Like AAV8, AAVBR1 has an early expression onset, although it significantly exceeds expression levels of both AAV8 and AAV2 in RPE and the retina for at least six weeks post-injection. We found that a single dose of AAVBR1.GFP (5 × 10^8^ vg), representing the low–moderate range of subretinal dosage in mice, was sufficient to disperse capsid beyond the subretinal bleb to all retinal quadrants. The AAVBR1 capsid differs from AAV2 by only seven consecutive amino acids in its peptide-binding motif, conferring a considerable shift in its tropism profile [6,7]. The high affinity of AAV2 for heparan sulfate proteoglycans [11] ensures that AAV2 will rapidly bind wherever it lands within the heparin-rich subretinal landscape, vastly limiting its viral distribution following subretinal delivery. We suspect that enhanced dispersal of AAVBR1 is due to the 7AA substitution altering AAVBR1 target preference from heparin to an alternative cell surface moiety. Being less ‘sticky’ within the heparin-abundant space allows the AAVBR1 capsid to disperse further distances from the injection site before cell capture. The loss of AAVBR1 binding to heparin is supported by multiple studies showing negligible capture of AAVBR1 by heparin-producing organs such as liver and lung following systemic delivery [6,12,13,14]. In contrast, systemically delivered AAV2 is almost exclusively trapped by the liver. Co-injecting a 500× molar excess of the unique AAVBR1 capsid peptide with AAVBR1.GFP or with AAV2.GFP significantly abrogated the signal in AAVBR1-injected eyes, but had no effect on AAV2 binding and expression, suggesting that the peptide specifically competes with AAVBR1 and not AAV2 for binding sites. However, it must be stated that the small sample size of the AAV2 and AAV2 + peptide groups, limits the robustness of this result. Moreover, the overall poor transduction capacity of AAV2 may diminish the precision of measuring interference from AAVBR1 peptide competition.

While the greater immune privilege of the subretinal space remains an important consideration for AAV-based ocular therapy [15,16,17], recent studies have confirmed that AAV-mediated photoreceptor and RPE toxicity can be associated with dosage and AAV cis-regulatory sequences [18,19], underscoring the urgency of minimizing therapeutic dosage while maximizing targeting efficiency. We constructed our AAVBR1.GFP expression plasmid using the universal CMV promoter. However, promoter engineering can be easily used to limit vector expression to specific cell types. The high transduction potency of AAVBR1 following subretinal delivery, combined with cell- or tissue-specific expression, is expected to further enhance the therapeutic applications of AAVBR1, which remains the only AAV capsid capable of targeting retinal (and CNS) microvasculature.

## Figures and Tables

**Figure 1 ijms-23-07738-f001:**
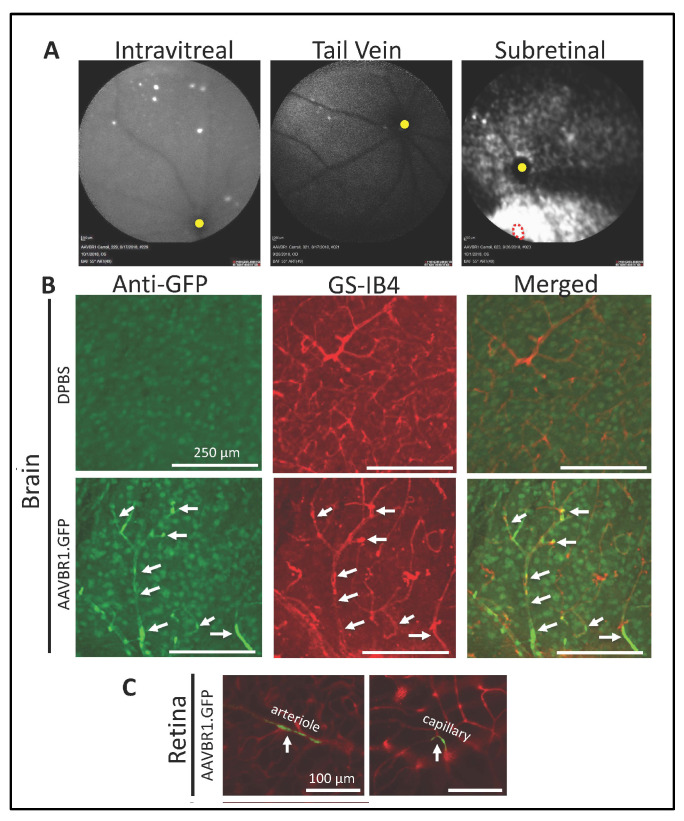
Fundus AAVBR1 expression depends on delivery route. (**A**). Fundus images one month after injection of AAVBR1.GFP. Intravitreal and systemic delivery show minimal retinal GFP fluorescence, whereas substantial signal is apparent following subretinal delivery. Optic disks are indicated by yellow dot. Subretinal injection site is indicated by dotted red line. (**B**). Colocalization of AAVBR1.GFP and GS-IB_4_ signals in brain vasculature (white arrows) following systemic injection of AAVBR1.GFP, with only background autofluorescence seen in the brain of a systemic DPBS-injected mouse (**C**). Systemic injection yields only minimal signal in retinal vasculature, indicated by GFP/GS-IB_4_ colocalization.

**Figure 2 ijms-23-07738-f002:**
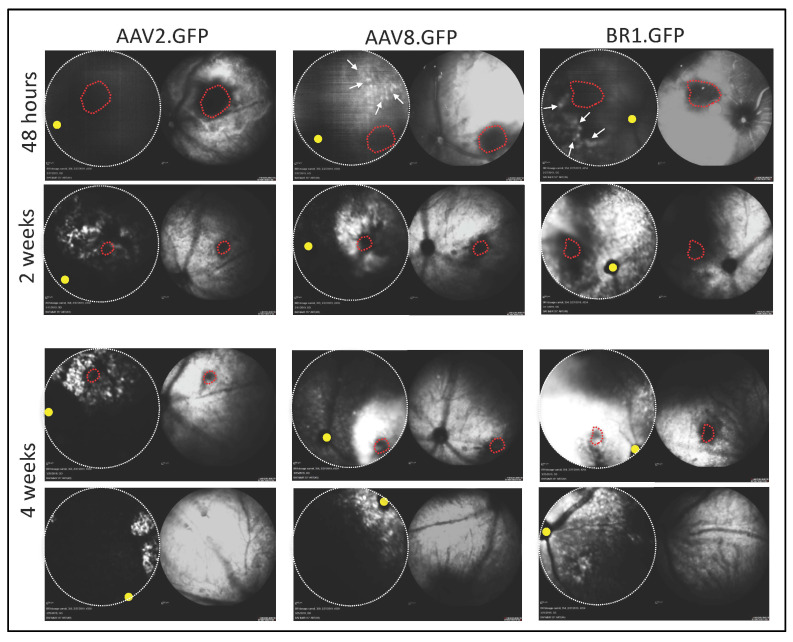
AAVBR1.GFP shows early and widespread expression in vivo following subretinal injection. Representative Spectralis images of three mouse retinas injected with either AAV2.GFP, AAV8.GFP, or AAVBR1.GFP (5 × 10^8^ vg) and imaged at 48 h, 2 weeks, and 4 weeks. **Left images** show fundus GFP signal using short-wavelength autofluorescence (FA mode). Corresponding **right** infrared **images** confirm orientation. Dotted red line indicates injection site. Yellow dot indicates optic disk. By 48 h post-delivery, faint in vivo GFP signal is visible in both AAVBR1.GFP- and AAV8.GFP-injected retinas (indicated by arrows), whereas AAV2.AcGFP expression is undetectable at this early timepoint. Four weeks after injection, widespread expression of AAVBR1.GFP is seen both proximal (**second to bottom panels**) and distal (**bottom panels**) to the injection site, compared to limited lateral spread of GFP signal following AAV2 or AAV8 delivery.

**Figure 3 ijms-23-07738-f003:**
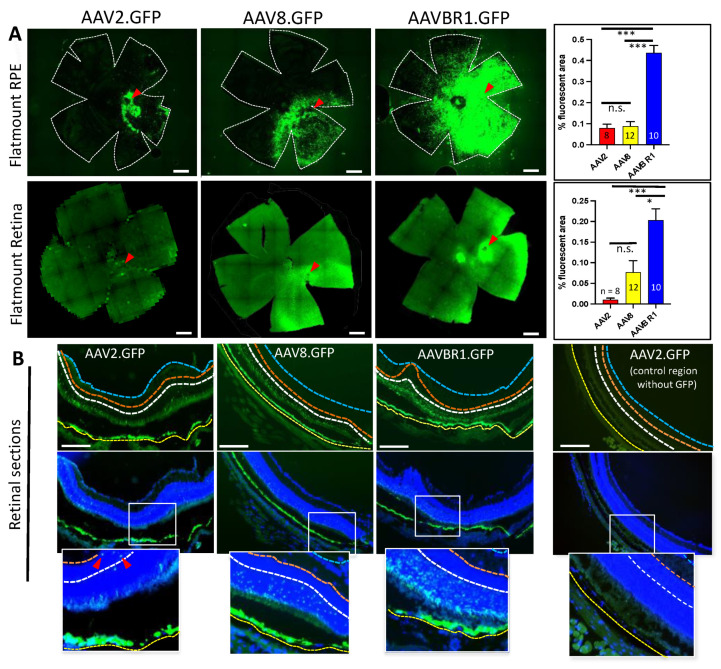
Six-week post-injection comparison of GFP signal following subretinal injection of 5 × 10^8^ vg AAV2, AAV8, or AAVBR1. (**A**). Top panels show representative images and corresponding bar graphs of average % fluorescent area in RPE. Bottom panels are retinal images with corresponding bar graphs from the same eye as RPE panels above. Numbers of eyes examined are indicated in graphs (scale bar = 500 μm, error bars represent SEM, *p* < 0.001 ***; *p* < 0.05 *). Red arrowheads show approximate injection sites. (**B**). Sections of AAV.GFP retina/RPE with and without DAPI nuclear stain. Considerable disparity in AAV2, AAV8, and AAVBR1 tropism for retina is apparent even in sections with nearly uniform GFP signal in underlying RPE. A section without GFP signal from the AAV2.GFP-injected eye is shown for comparison. Location of insets is indicated by white outlined rectangles. Red arrowheads point to GFP-positive cells within the inner nuclear layer of the AAV2-injected eye. Yellow, white, orange, and blue dotted lines indicate basal boundaries of RPE, outer plexiform layer (OPL), inner plexiform layer (IPL), and nerve fiber layer (NFL), respectively (scale bar = 200 μm).

**Figure 4 ijms-23-07738-f004:**
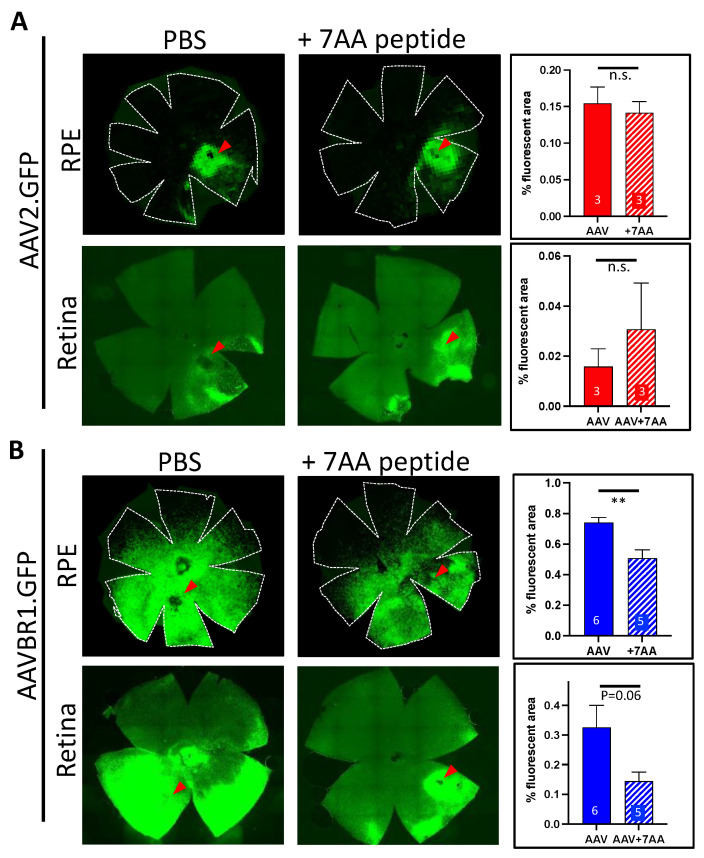
GFP signal from subretinal AAVBR1.GFP injection in eyes is considerably abrogated by co-injection of the 7AA peptide specific to the AAVBR1 capsid. Representative RPE and retinal flatmounts show that subretinal co-injection of 1:500 molar ratio of AAV.GFP:peptide has no effect on (**A**) AAV2-mediated signal, but significantly reduces signal in (**B**) AAVBR1-injected RPE at 10 weeks post-injection (*p* < 0.005 **). Corresponding graphs with % autofluorescence are shown in panels on the right, indicating number of eyes examined for each analysis. Red arrowheads show approximate injection sites. Numbers of eyes examined are indicated in graphs. Error bars represent SEM.

## Data Availability

The data presented in this study are contained within the Appendix A.

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
