# Peer review of "Tropism of the Novel AAVBR1 Capsid Following Subretinal Delivery"

_ijms, 2022, doi:10.3390/ijms23147738_

Round 1

Reviewer 1 Report

In the manuscript by Carroll et al. the authors describe increased transduction and lateral spread to the retina after subretinal injection of a capsid modified variant of AAV2 called AAVBR1. The authors use GFP marker gene delivery and compares the transduction profile in the eye for AAVBR1 to that of the parental AAV2 and another widely used serotype AAV8. They do so by fluorescens fundoscopy, staining of flatmounts and retinal cross sections.

While the study is highly interesting in order to determine the transduction capacity of this newly reported capsid variant (Korbelin et al. 2016), the authors fails to do so in a convincing manner.

Specific comments:

Figure 1: Control experiments fails to clearly demonstrate delivery to endothelial cells following systemic delivery. The legends are poor, and specific description of part A, B and C are missing. The staining protocol used for GS-IB4 is absent. For subretinal injection the injection site should be clearly marked.  

All GFP readouts show be depicted as green in the figures for easy presentation

Figure 2: Major issue as described below regarding the strength and validity of the data.

The eyes should be aligned across time point.

Please clarify what the IR images are telling

Figure 3:  Main data in this paper. See comment below (major issue)

RPE and retinal flarmounts should be oriented alike

Control experiment missing (non-GFP, vehicle)

Cross section of the whole eye would add value

Stain of nucleus would improve the analysis of part B

Key for { symbol is missing.  

Figure 4: Nice control experiment, but the effect on AAV2 is impossible to quantify as transduction capacity is limited by ‘default’, and since N er very low

The data for AAVBR1 folloows the same critique as for figure 3.

Major issues:

It is impossible to assess power and statistical validity of this study (in particular Fig 3 and 4) since a detailed description of how many animals were excluded, how many animal and eyes were quantified and so forth is missing. The total number of images/eyes and animals used are very low, especially given the fact that subretinal injections vary substantially from injection to injection, and from animal to animal. Given the small numbers of images analyzed here, all of them should be presented, at least as an appendix.

Pictures of the ‘bleb formation’ following injections should have been obtained, and clear marking of all injection sites (when appropriate) should have been added.

The authors argue that lateral spread is increased for BR1, and suspect that this occurs as heparin binding is lost. However, it makes little sense that a ‘blocking peptide’ then reverses this pattern. That would argue for a competitive binding to a target instead. Please clarify. The authors should substantiate their discussion in much greater detail, and look at the transduction profiles observed by others and by Korbelin et al recently. A control experiment using co-delivery of an irrelevant peptide might be worth pursuing.  

Minor:

Typos (as AAAV), differing font size, different font colors, inconsistent use of capitalization

Reviewer 2 Report

In this manuscript, the authors demonstrate that a CNS microvasculature endothelial cell-specific vector AAVBR1 has superior tropism to RPE and outer retina than either AAV2 or AAV8 following subretinal delivery. The topic is interesting and worth publishing. Below are suggestions to improve the manuscript:

The author only showed single staining of GFP in eye section (Figure 4B). Double staining of GFP with RPE marker (e.g. RPE-65) and DAPI is needed to localize GFP in the RPE layer and outer retina.

Author Response

We thank the reviewer for taking time to examine our manuscript and help us make it better! As both reviewers wanted us to include DAPI signal to confirm the GFP signal is infact RPE, we hope our new figure 3 is satisfactory. Our original images were actually co-stained with DAPI so we were able to add GFP/DAPI merged images to explicitly show that DAPI nuclei at the same level as GFP signal in RPE, particlularly in the mangified insets. We also added a section image of the AAV2.GFP injected eye at a region with NO GFP signal to further convince readers that the RPE signal is not due to autofluorescence.

Round 2

Reviewer 1 Report

Dear authors,

I believe most of the concerns raised in the previous draft was addressed in the revised version, and I find it acceptable for publication 

Author Response

Dear reviewer,

We very much appreciate your careful reading of our manuscript and your suggestions to make our  manuscript more transparent and publication-ready. We are happy that you found the revision to be acceptable for publication!